# Standardization of Data Analysis for RT-QuIC-Based Detection of Chronic Wasting Disease

**DOI:** 10.3390/pathogens12020309

**Published:** 2023-02-13

**Authors:** Gage R. Rowden, Catalina Picasso-Risso, Manci Li, Marc D. Schwabenlander, Tiffany M. Wolf, Peter A. Larsen

**Affiliations:** 1Department of Veterinary and Biomedical Sciences, University of Minnesota, St. Paul, MN 55108, USA; 2Minnesota Center for Prion Research and Outreach, University of Minnesota, St. Paul, MN 55108, USA; 3Department of Veterinary Population Medicine, University of Minnesota, St. Paul, MN 55108, USA

**Keywords:** RT-QuIC, real-time quaking-induced conversion, ELISA, diagnostics, chronic wasting disease, maxpoint ratio, prion

## Abstract

Chronic wasting disease (CWD) is a disease affecting cervids and is caused by prions accumulating as pathogenic fibrils in lymphoid tissue and the central nervous system. Approaches for detecting CWD prions historically relied on antibody-based assays. However, recent advancements in protein amplification technology provided the foundation for a new class of CWD diagnostic tools. In particular, real-time quaking-induced conversion (RT-QuIC) has rapidly become a feasible option for CWD diagnosis. Despite its increased usage for CWD-focused research, there lacks a consensus regarding the interpretation of RT-QuIC data for diagnostic purposes. It is imperative then to identify a standardized and replicable method for determining CWD status from RT-QuIC data. Here, we assessed variables that could impact RT-QuIC results and explored the use of maxpoint ratios (maximumRFU/backgroundRFU) to improve the consistency of RT-QuIC analysis. We examined a variety of statistical analyses to retrospectively analyze CWD status based on RT-QuIC and ELISA results from 668 white-tailed deer lymph nodes. Our results revealed an MPR threshold of 2.0 for determining the rate of amyloid formation, and MPR analysis showed excellent agreement with independent ELISA results. These findings suggest that the use of MPR is a statistically viable option for normalizing between RT-QuIC experiments and defining CWD status.

## 1. Introduction

The diagnosis of prion diseases has historically relied on immunoassays such as enzyme-linked immunosorbent assay (ELISA) and immunohistochemistry (IHC) [1]. However, recent advancements in protein amplification assays such as real-time quaking induced conversion (RT-QuIC) have emerged as powerful tools for detecting a variety of transmissible spongiform encephalopathies (TSEs) and other proteopathies, including chronic wasting disease (CWD), scrapie, bovine spongiform encephalopathy (BSE), Creutzfeldt–Jakob disease (CJD), Alzheimer’s disease, and Parkinson’s disease [2,3,4,5,6,7,8,9]. CWD is caused by the misfolding of prion protein (PrP${}^{CWD}$) and the subsequent recruitment of native prion protein (PrP${}^{C}$) into pathological amyloid fibrils, thus causing a contagious and unconditionally fatal disease [10,11]. The disease was first described in captive mule deer in Colorado in 1967 and has since been identified across the United States, Canada, Scandinavia, and South Korea [12,13,14]. Because of the severity and continued spread of the disease, diagnostic methods must evolve to meet the growing need for fast and accurate testing. RT-QuIC quantifies amyloid fibril formation via relative fluorescence (RFU) using thioflavin T (ThT) as a fluorescent marker and has shown similar-to-improved accuracy when compared to IHC and ELISA [15,16]. Moreover, the United States Department of Agriculture Animal and Plant Health Inspection Service (USDA APHIS) is currently validating RT-QuIC for the routine detection of CWD in farmed cervids (Tracy Nichols; personal communication).

Despite the growing interest in RT-QuIC-based diagnostics, there is currently no consensus regarding the interpretation of the assay’s output. Most authors determine CWD positivity by setting a predefined number of standard deviations (e.g., between 3 and 30) above the average background ThT fluorescence per plate and defining that fluorescent value as a positivity threshold (T${}_{stdev}$). A rate of amyloid formation (RAF) is then calculated as the inverse of the time that the reaction took to pass T${}_{stdev}$. Finally, a positive sample is determined arbitrarily when some minimum percentage (generally between 25% and 67%) of replicates passes T${}_{stdev}$ [1,3,4,5,6,7,17,18,19,20,21,22,23,24,25,26,27,28,29,30,31,32,33,34,35,36,37,38,39,40,41,42,43,44,45,46,47,48,49,50,51,52,53,54,55,56,57] (Table 1). Additionally, some researchers incorporate statistical analyses, such as Mann–Whitney, Wilcoxon signed-rank, or Fisher’s exact test, which statistically compares the RAF replicates of samples with an unknown CWD status to those of a control sample [1,18,27,38,52,53,55]. Although RAFs can provide a quantitative assessment of relative PrP${}^{CWD}$ load, they are not statistically informative when compared to an ideal negative control, and CWD determinations from their analyses are ultimately susceptible to researcher preference. This is because, for an ideal negative control, all RAF values are zero, which is not statistically useful, whereas the MPR values for the same sample will have some variability, typically ranging from one to two. This challenges standardization among laboratories and the external validity of results. Given the increased usage of RT-QuIC for CWD research and surveillance, a robust assessment of the analytical process to define disease status with RT-QuIC data is needed.

Here, we describe a method for determining the CWD status of a given sample based on what we coined the maxpoint ratio (MPR) [58]. This method was adapted from Cheng et al. [59,60] and corrects replicates for background fluorescence and normalizes data output between experiments. CWD-positivity determinations using MPR are based on a statistical analysis against the MPR values of a known negative control rather than deciding beforehand the necessary number of replicates needed to cross a threshold. Nevertheless, the RAF of a reaction is still an important kinetic measurement that gives an idea of the relative prion load in a sample, so a constant threshold (T${}_{MPR}$) was determined at twice the background fluorescence for RAF calculations rather than assigning independent thresholds (T${}_{stdev}$) per plate. This threshold was determined using ROC analysis against tissue-matched ELISA results. However, independent analysis by other laboratories may indicate a different threshold. For this reason, independent, empirical confirmation T${}_{MPR}$ should be determined per laboratory. With the use of this proposed method, we aim to normalize the results across RT-QuIC experiments, thus standardizing the RT-QuIC diagnostic results across laboratories and improving CWD control programs.

## 2. Results

### 2.1. Literature Review

The literature review conducted in PubMed identified 46 peer-reviewed publications fitting our search criteria (see Table 1) (keywords: RT-QuIC and “real-time quaking-induced conversion”; between the years 2012 and 2021). Upon the inspection of the methods used to analyze RT-QuIC data, we discerned a common theme for defining a positivity threshold (T${}_{stdev}$). Notably, 43 studies averaged the initial fluorescence of every well on a reaction plate and defined T${}_{stdev}$ as some number of standard deviations (between 3 and 30) above that average. One study [24] defined T${}_{stdev}$ as 33% of the maximum fluorescence of the strongest positive control. Another [48] defined T${}_{stdev}$ as 10,000 relative fluorescent units (RFU), while another [51] defined it as 120 RFU. Once T${}_{stdev}$ was determined, three approaches were typically adopted to determine the positivity for a particular proteopathy: (1) some number or percentage of replicates crossing T${}_{stdev}$ (*n* = 30), (2) comparison to a negative control using a statistical test (*n* = 11), or (3) the average RFU of a sample’s replicates was above T${}_{stdev}$ (*n* = 5). The analytical approach used for CWD diagnosis was highly variable, without any clear agreement in RT-QuIC analysis regardless of tissue type.

### 2.2. Maxpoint Ratio, Rate of Amyloid Formation, and Time-to-Threshold Distributions

MPR values of 4065 individual RT-QuIC replicates from 46 reaction plates were compiled, and two distinct distributions were observed (Figure 1A). Once the distributions were determined, all control replicates were removed to account for any bias for those samples. MPRs <2.0 comprised the entirety of the lower distribution. Given this observation, we established a T${}_{MPR}$ at an MPR value of 2.0 for calculating RAF. A density curve was fitted to the lower MPR distribution (*n* = 2974; R${}^{2}$: 0.9980; *p*: 0.6270; Figure 1B). For the upper MPR distribution, values were distributed normally, although approximately 20% of the variance was unexplained by the model (*n* = 490; R${}^{2}$: 0.8464; *p*: 0.1039; Figure 1C). However, those replicates’ time-to-threshold (TtT) (R${}^{2}$: 0.9873; *p*: 0.6270) and RAF values (R${}^{2}$: 0.9708; *p*: 0.1136) were both well supported by their lognormal regression models (Figure 1D,E). The same analyses were performed on the isolated control values (Figure 2B). The pattern for negative-control MPRs matched that of the lower distribution mentioned above (*n* = 285; R${}^{2}$: 0.9993; *p*: 0.0513; Figure 2A). Positive-control MPRs were distributed normally, but approximately 10% of the variance was unexplained by the regression (*n* = 316; R${}^{2}$: 0.8985; *p*: 0.2137; Figure 2C). However, the lognormal regression model for positive control RAFs was an excellent fit (*n* = 316; R${}^{2}$: 0.9912; *p*: 0.5860; Figure 2D). The low p-values for both Figure 1B and Figure 2A indicate that these distributions are positively skewed.

### 2.3. T${}_{MPR}$ and Rate of Amyloid Formation Determinations

A conservative threshold (T${}_{MPR}$) was set at a value of 2.0 for determining RAF with the MPR method. This T${}_{MPR}$ was based on the distributions highlighted in Figure 1. The example graphs showing the output of this method are demonstrated in Appendix A. A receiver operating characteristic (ROC) analysis was performed on individual MPR values for which there were tissue-matched, bilaterally sampled ELISA results (Figure 3). The analysis revealed an area under the curve (AUC) of 0.9226 (95% CI: 0.8853–0.9600), suggesting that MPR has a diagnostic accuracy of 92.26% in comparison to ELISA. Cutoff values were then sorted in descending order based on Youden’s index values (*J*), which indicate the best tradeoff between sensitivity and specificity. The two highest indices were obtained with MPR values of 1.957 and 2.523. These support the T${}_{MPR}$ value selected, given there is no substantial difference in performance (Se and Sp) between both highest indices, and the range includes the value of 2. Notably, five individual WTD had RT-QuIC replicates >2.0 but were negative for ELISA. Two of these samples were detected as CWD-positive by ELISA in medial retropharyngeal lymph nodes (RPLNs) [1] but not in the tissue matched with these RT-QuIC results.

### 2.4. Distribution Differences between Lymph Nodes

When the lower distribution was further divided based on tissue type, parotid (PLN; *n* = 2313, R${}^{2}$: 0.9993; *p*: 0.0736), submandibular (SMLN; *n* = 217; R${}^{2}$: 0.9982; *p*: 0.1936), and tonsil lymph node (TLN; *n* = 427; R${}^{2}$: 0.9919; *p*: 0.4846) distributions were all still very well explained by the normal regression model (Figure 4A). The *p*-values for all of the density plots were fairly low, indicating that they all skewed in the positive direction. The multiple-comparison analysis showed that there was no significant difference between any of the tissue-specific distributions (Figure 4B).

### 2.5. Correlation between MPR and RAF

RAFs were plotted against their corresponding MPRs to determine any relationship between the two (Figure 5). Spearman’s coefficient (*r*: 0.5123, 95% CI: 0.4578 to 0.5629) suggests that they are mostly independent of each other. The weak correlation resulted in a poor fit of the linear regression as well (R${}^{2}$: 0.2366).

### 2.6. Machine and Batch Effects on MPR and RAF

We examined the variability of MPRs and RAFs between plate readers and recombinant substrate (recPrP) batches to understand their potential effects on the assay. We detected no significant differences in negative-control MPRs between the batches (Figure 6A). However, significant variability was observed in those MPRs based on plate readers (*p*: <0.0001–0.0129), except for one comparison (1 vs. 4; *p*: 0.9995; Figure 6B). The only significant batch effects for the positive-control MPRs were noticed in batches 1 vs. 3 (*p*: 0.0004) and 3 vs. 4 (*p*: 0.0242; Figure 6C). We detected significant differences between positive-control MPRs produced by plate readers 1 vs. 3 (*p*: 0.0007) and 2 vs. 3 (*p*: < 0.0001; Figure 6D). Notably, the first two batches of recPrP (1 and 2) yielded significantly slower RAFs than subsequent batches (3 and 4) (*p*: < 0.0001; Figure 6E). Significant differences were detected in RAFs between all plate readers (*p*: < 0.0001–0.0321) except 1 vs. 2 (*p*: 0.9612) and 3 vs. 4 (*p*: 0.9999; Figure 6F).

### 2.7. Comparison of RT-QuIC to ELISA

Because ELISA is used as the standard screening process for CWD diagnosis, a comparison was necessary to determine the agreement between MPR results and ELISA results. Out of the 420 RT-QuIC replicates produced from 64 lymph node samples (with one sample tested twice due to a conflicting result) for which tissue-matched ELISA data were available, 402 replicates agreed with the ELISA results, whereas 18 replicates disagreed (κ: 0.863, 95% CI: 0.801–0.924; Figure 7A). After comparing MPR replicates to their respective plates’ negative controls (such as in Appendix A) using ANOVA and Dunnett’s multiple-comparison test, 60 samples agreed with ELISA, 5 samples disagreed with an ELISA-negative result, and no samples disagreed with an ELISA-positive result (κ: 0.780, 95% CI: 0.599–0.961; Figure 7B). Of the five WTD whose results disagreed with the ELISA-negative result for a certain tissue type, two were confirmed CWD-positive in a different tissue type by ELISA [1].

## 3. Discussion

Currently, there is no standardized, statistical approach for determining prion disease status when utilizing RT-QuIC. The most commonly used method is based on T${}_{stdev}$, the calculation of which involves acquiring a baseline mean RFU of an entire reaction plate and then determining a T${}_{stdev}$ as a varying integer of standard deviations from that value. RAFs are then defined as the inverse of the time taken to reach the T${}_{stdev}$; however, the magnitude of these values is typically not considered in diagnostic determinations. Importantly, this method does not take into account the initial variation between replicates, samples, or particular experiments. Thus, reliably comparing multiple experiments becomes increasingly complex. Furthermore, meaningful statistics are difficult to perform on the RAFs alone when there is no variance in an ideal negative control (i.e., all replicates have an RAF of zero). In such cases, using a *t*-test or Mann–Whitney test is not statistically informative because a positive is determined solely based on how many wells cross T${}_{stdev}$. Just as with RAFs calculated through this approach, those obtained by crossing T${}_{MPR}$ have identical statistical limitations. Therefore, we believe that RAF values, regardless of provenance, are more useful for understanding the seeding kinetics of a given sample than for determining a diagnosis.

Using MPRs instead solves the statistical issues associated with interpreting RT-QuIC data with RAF values alone. Of 2974 negative sample replicates from over 500 individuals, MPR values were distributed normally (see Figure 1). Further, when the lower distribution was separated based on tissue type, no significant difference was detected (see Figure 4). This means that when comparing an unknown sample to a known negative lymph node sample, researchers can perform a meaningful, parametric analysis that does not rely on arbitrary determinations. While a tissue-matched negative sample is ideal, we also show in Figure 5 that between the negative PLNs, SMLNs, and TLNs, there was no significant difference between their MPR distributions. Therefore, we are confident that using one control lymph node is sufficient for comparison at least between these three tissue types. Further study on negative samples of other tissue types needs to be conducted to determine if this pattern remains. The SMLN distribution (Figure 4, *n* = 217) also shows that the dataset does not need to be exceedingly large for researchers to obtain an accurate distribution estimate of a negative control. It should be noted that this study focused only on lymphoid tissue, so MPR distributions for other tissue types should be independently determined.

Interestingly, there is seemingly no correlation between individual replicates’ MPR and RAF values, indicating that the two measurements are mostly independent of each other (Figure 5). This may be due to the observation that MPRs for a single tissue sample (in this case, the positive control) have a more weakly supported distribution model (Figure 2C); nevertheless, the RAF values for the same samples distribute in a highly supported, normal model (Figure 2D). Further, one would expect MPR values between any given sample to always be similar because the recPrP concentration is identical for every reaction. This of course assumes that all recPrP is consumed during the run and that fibril formation is homogeneous. However, our recPrP batch comparisons (Figure 6) revealed that this assumption of consistency does not necessarily uphold, thus implying that the reaction kinetics of RT-QuIC are more complex.

The most notable effect on the negative-control MPR distribution occurs between the plate readers (Figure 6B). Despite running the same program script, there appeared to be some amount of variation due to machine effects. It is worth noting that microplate reader #1 was an earlier model developed by BMG and was not designed specifically for RT-QuIC. This model lacks plate stabilizers, which reduce unintended, sporadic vibrations to the plate. Nevertheless, significant differences were found between all readers (except 1 vs. 4). For the positive-control MPR distribution, some difference was found in batch 3, although this may be due to the small sample size (*n* = 20) for that batch (Figure 6C). Additionally, reader 3 showed some significant differences from the other three readers; however, the range of these values was still well within the other three (Figure 6D). RAF values were more sensitive to recPrP batches. Because the rate significantly increased with newer recPrP batches (batches 3 and 4), this effect may be explained by the improved proficiency of our team with the purification method. Furthermore, these machine and batch effects may contribute to some of the variations not explained by the regression models, and further research needs to be conducted to determine this.

MPR distributions may be influenced by a multitude of variables, such as reader settings, recombinant substrate, tissue type, temperature, and fluorescent filters. Therefore, we suggest that T${}_{MPR}$ for RAF calculations be determined by researchers empirically to account for these variables, although a tentative T${}_{MPR}$ of 2.0 is conservative given that it is roughly 9.54 standard deviations from the average of the distribution shown herein (Figure 1). Further, this cutoff was also supported by ROC analysis, where MPRs were compared with tissue-matched ELISA results. Notably, two of the five tissue samples that were negative for ELISA but positive for RT-QuIC came from known CWD-positive animals, suggesting that RT-QuIC is more sensitive than ELISA. This observation would have artificially skewed the ROC analysis to indicate a lower diagnostic specificity for RT-QuIC than is actually real. Further, the gain setting, which amplifies the fluorescent signal to the sensor, is important for acquiring accurate MPR measurements; if a replicate saturates the fluorescent sensor, the actual MPR value is unknown and will skew any statistical analysis. The gain (typically set between 1000 and 1600) can be adjusted appropriately to account for this. Lastly, because all RT-QuIC runs were performed at 42 °C, it is unknown how temperature would affect the observations shown herein.

A unique characteristic of this method is that replicates do not necessarily need to cross T${}_{MPR}$ to be determined as significantly different from the negative control using MPR. A clear example of this would be if a sample’s MPR replicates are significantly lower than the replicates of the negative control. However, if a sample’s distribution is <2.0 but is significantly different from the negative control, we suggest retesting the sample. Notably, ThT fluorescence increases at lower temperatures, so the background measurement should never be taken from the first few minutes of the experiment. The reaction must equilibrate to the experimental temperature to determine an accurate background value. We recommend an equilibration time of at least 45 min.

We propose that using a combination of MPR and RAF allows for a more robust analysis of RT-QuIC data. Across the 4065 replicates, which were run on 4 separate plate readers and 4 separate recPrP batches, negative MPRs were distributed normally (see Figure 1). Furthermore, statistical analysis of 64 samples showed that there was an excellent agreement with independently secured ELISA results (κ: 0.780), demonstrating that by using a ratio based on a replicate’s independent background fluorescence, RT-QuIC reactions can be normalized between experiments, plate readers, and recPrP batches. Given the expanded usage of RT-QuIC for CWD and other proteopathies and the anticipated approval for regulated testing of CWD in the United States, it is imperative to have a standardized, statistical method that accounts for multiple variables across experiments. We propose that the MPR method presented herein, at least in part, fulfills that qualification.

## 4. Materials and Methods

### 4.1. Literature Review

To identify the methods used by researchers to determine TSE status from RT-QuIC data, we conducted a PubMed search (keywords: RT-QuIC and “real-time quaking-induced conversion”) for articles that performed RT-QuIC between 2012 and 2021. Studies were selected if the researchers both performed RT-QuIC and implemented a rubric for determining TSE status. Studies were not vetted for any particular prion disease or proteopathy.

### 4.2. Source Population and Sample Processing

In accordance with the culling efforts of the Minnesota Department of Natural Resources (DNR), 668 lymph node samples from 533 wild white-tailed deer (WTD) were sampled from CWD endemic areas in southwestern Minnesota [1]. Following the DNR’s CWD surveillance program mentioned above, pooled homogenates of three subsamples for each animal’s RPLNs were submitted for independent screening by ELISA to the Colorado State University Veterinary Diagnostic Laboratory (CSU VDL, Fort Collins, CO, USA). Thirteen positive RPLN samples were identified by CSU and were returned to the Minnesota Center for Prion Research and Outreach (MNPRO; see below) for RT-QuIC testing. Unfortunately, any ELISA-negative RPLNs were disposed o before being able to be tested by RT-QuIC.

Additionally, parotid (PLN; *n* = 515), CWD-positive medial retropharyngeal (RPLN, *n* = 17), submandibular (SMLN; *n* = 63), and palatine tonsil (TLN; *n* = 62) lymph nodes were collected and stored at −20 °C. For controls, PLNs from known CWD^+^ and CWD^−^ WTD were used. Approximately 100 mg of lymph node tissue were homogenized in 1X PBS to a concentration of 10% (*w*:*v*) using a BeadBug™ homogenizer (Benchmark Scientific, Sayreville NJ, USA) and 1.5 mm zirconium beads (Millipore Sigma, Burlington, MA, USA) for 90 s at the maximum setting. Homogenates were stored at −80 °C until tested, at which point they were diluted 100-fold (10-3 final dilution) in dilution buffer (1X PBS, 0.1% (*w*:*v*) sodium dodecyl sulfate, 1X N${}_{2}$ supplement (Life Technologies Corporation, Carlsbad, CA, USA)) and were vortexed [1]. RT-QuIC was performed at MNPRO.

Following Schwabenlander, et al. (2021) [1], a subset of PLNs, SMLNs, and TLNs from 60 WTD were selected for blind testing by RT-QuIC at MNPRO. CSU VDL then conducted ELISAs on PLNs, SMLNs, and TLNs that were detected by RT-QuIC to determine concordance between the two assays. ELISAs were performed using the Bio-Rad TeSeE Short Assay Protocol (SAP) Combo Kit (BioRad Laboratories Inc., Hercules, CA, USA). Importantly, while the Bio-Rad ELISA is used as the standard screening assay for CWD, it is only currently validated for RPLNs and obex.

### 4.3. RT-QuIC Methods and recPrP Preparation

The expression and purification of recombinant prion protein (recPrP; hamster, amino acids 90-231) and RT-QuIC assays were performed following Schwabenlander et al. [1]. Briefly, 98 μL of the reaction master mix and 2 μL of the 10${}^{-3}$ homogenate in dilution buffer were added to each well of a 96-well plate. The final tissue concentration was 0.002% (*w*:*v*). Reactions were performed in quadruplicate for all the samples (*n* = 668), and six replicates per plate for positive and negative controls. All the experiments were run using four BMG FluoStar® plate readers (BMG Labtech, Ortenberg, Germany). Assays were performed at 42 °C, and readings were taken every 45 min with one-minute cycles of double-orbital shaking and resting. The gain was set to 1600. Four different batches of recPrP substrate were used in the span of this study and were produced at The University of Minnesota Biotechnology Resource Center.

### 4.4. Maxpoint Ratio and Rate of Amyloid Formation Calculations

MPRs and their associated RAF values were adapted from the algorithm of Cheng, et al. [59]. MPRs were calculated by dividing the maximum RFU value obtained within 48 h by the background RFU value for each individual replicate (i.e., MPR = maxRFU/backgroundRFU). MPRs from each replicate were plotted using GraphPad Prism version 9.0.1 (GraphPad Software, San Diego, CA, USA) to determine any observable distribution patterns (Figure 1A). A constant T${}_{MPR}$ of 2.0 was chosen as a convenient yet empirical cutoff for determining RAF. A formal assessment using ROC analysis to support the empirical threshold of (T${}_{MPR}$) was performed by comparing individual MPR values to tissue-matched ELISA results (Figure 3). Cutoff values were listed in descending order of Youden’s index (*J* = sensitivity + specificity − 1). RAFs were computed retrospectively by taking the reciprocal of the time in seconds needed to cross T${}_{MPR}$. If a replicate did not cross T${}_{MPR}$ within 48 h, it was assigned an RAF of zero (see Appendix A).

### 4.5. Statistical Analysis

Data curation and management were performed in the R software v4.1.0 (R Core Team 2021), and GraphPad Prism version 9.0.1 (GraphPad Software, San Diego, CA, USA) was used for all statistical analyses. The relative frequency distributions and density plots for MPR values were constructed for both unknown (*n* = 3560) and control (*n* = 601) samples. For those replicates which crossed T${}_{MPR}$, frequency distributions, and density plots were constructed for the TtT and RAFs for unknown (*n* = 524) and control (*n* = 316) replicates. R${}^{2}$ values and *p*-values were calculated to diagnose the regressions (D’Agostino–Pearson omnibus [K2] normality of residuals test). A low *p*-value indicates that the distribution is skewed, and an R${}^{2}$ near 1 suggests that the model explains closer to 100% of the data variance. The lower MPR distribution was further divided based on three tissue types: PLN (*n* = 2594), SMLN (*n* = 217), or TLN (*n* = 427). Due to insufficient data points, RPLN (*n* = 17), PPLN (*n* = 30), and PSLN (*n* = 16) replicates were excluded from this analysis. The differences in tissue-specific MPR distributions were compared using a Brown–Forsythe and Welch ANOVA, and a Dunnett’s T3 multiple-comparison test.

Control sample MPRs and RAFs were compiled to identify the potential variability introduced by a particular microplate reader and/or recPrP production batch used for a given experiment, and outliers were eliminated as mentioned above. A Brown–Forsythe and Welch ANOVA and a Dunnett’s T3 multiple-comparison test were used to determine the significance between readers or batches.

RAF values greater than zero (*n* = 850) were plotted against their corresponding MPR values to determine the relationship between conversion efficiency and ThT fluorescent intensity. Spearman’s rank correlation test was performed to determine the correlation between the individual replicate measurements.

CWD diagnosis using the MPR method was performed by comparing four MPR replicates to six replicates of each experiment’s corresponding negative control using a one-way ANOVA and a Dunnett’s multiple-comparison test (Figure 7B, Appendix A). A sample was considered positive when *p* < 0.05.

RT-QuIC results from the MPR method were compared with ELISA results when available. In total, 420 replicates from 64 samples were analyzed using Cohen’s kappa test to determine the reliability between ELISA and RT-QuIC results using the MPR method.

## Figures and Tables

**Figure 1 pathogens-12-00309-f001:**
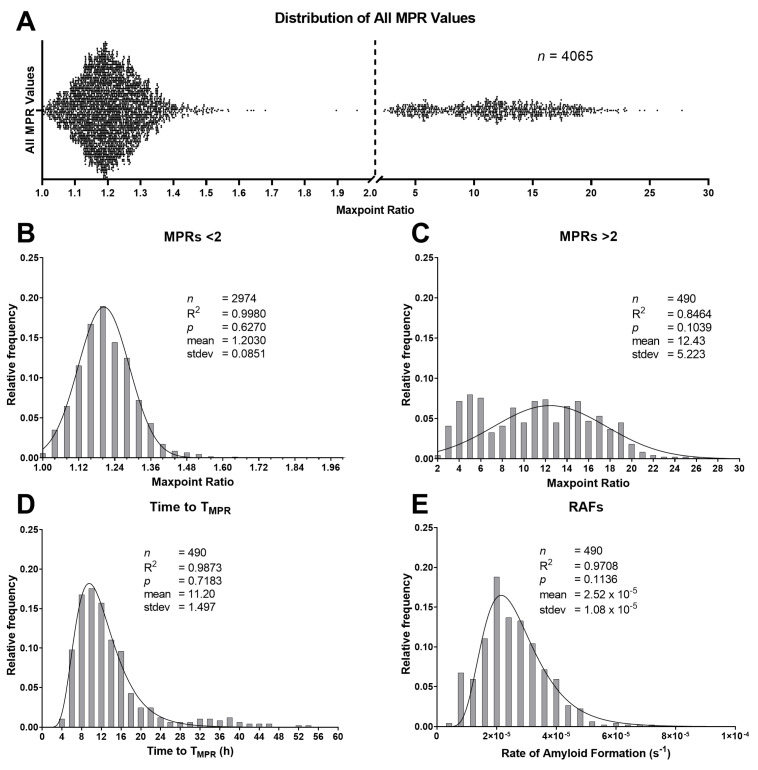
(**A**) All maxpoint ratios (MPR) showing two distinct distributions separated by the dotted line at 2.0; (**B**) the frequency distribution of replicates with MPR values <2.0. Tissue types included are parotid (PLN; *n* = 2313), submandibular (SMLN; *n* = 217), tonsil (TLN; *n* = 427), and retropharyngeal (RPLN; *n* = 17). It does not include negative controls (*n* = 285). These MPRs exhibited a normal distribution; (**C**) distribution of the MPRs >2.0 minus the control (*n* = 316) MPRs (PLN (*n* = 155), SMLN (*n* = 51), TLN (*n* = 101), and RPLN (*n* = 183)). The data were distributed normally although 15% of the variance was unexplained by the regression model; (**D**,**E**) time-to-threshold values and their corresponding rates of amyloid formation (RAF) were distributed in a lognormal pattern. stdev: standard deviation.

**Figure 2 pathogens-12-00309-f002:**
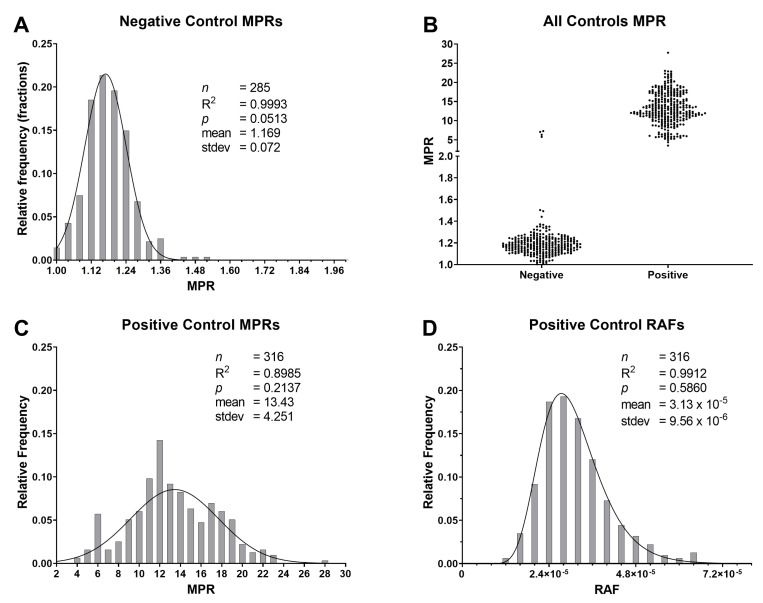
Isolated maxpoint ratios (MPR) for the RT-QuIC control samples: (**A**) negative-control MPRs fit a normal distribution and the values are slightly skewed in the positive direction; (**B**) distribution comparisons of the negative and positive controls; (**C**) the MPRs of positive controls fit a normal distribution; however, roughly 10% of the variance was unexplained by the model; (**D**) a lognormal distribution was observed for the corresponding rate of amyloid formation values of the positive control with an excellent fit for these values.

**Figure 3 pathogens-12-00309-f003:**
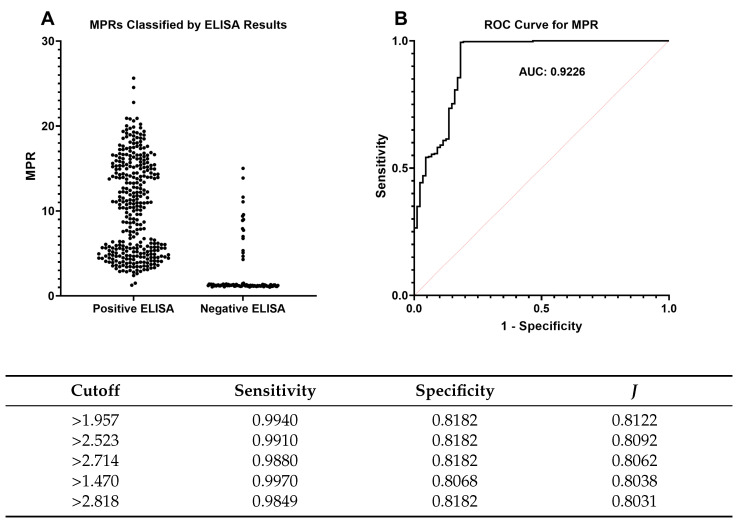
(**A**) Maxpoint ratios (MPR) of unknown-CWD-status sample replicates compared to tissue-matched, bilaterally-sampled, enzyme-linked immunosorbent assay (ELISA) results; (**B**) receiver operating characteristic (ROC) curve and its area under the curve (AUC; 0.9226) for individual replicates’ MPR vs. ELISA results. The table shows cutoff values with the highest corresponding Youden’s index (*J* = sensitivity + specificity − 1) in descending order. The top cutoff value supported our chosen T${}_{MPR}$ of 2.0 for determining the rate of amyloid formation (RAF).

**Figure 4 pathogens-12-00309-f004:**
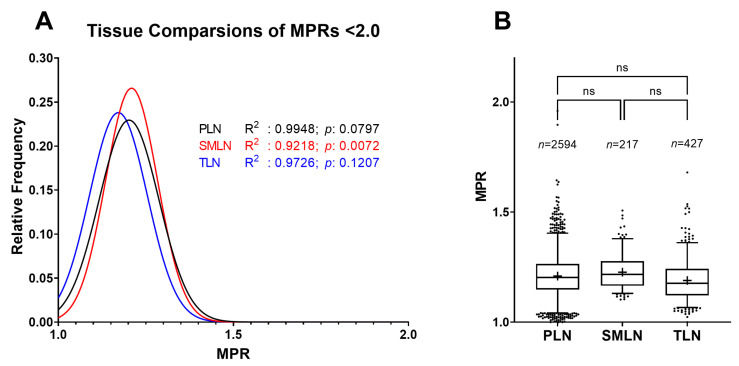
(**A**) Comparison of MPRs in the lower distribution separated by tissue type. Medial retropharyngeal lymph node values were excluded due to insufficient data (*n* = 17). Regression analysis supported the findings of the models regardless of the three tissue types; (**B**) the multiple-comparison test found no significance between any of the tissue-specific distributions. Boxes represent the interquartile range, lines are set at the median, “+’s” denote the mean, whiskers are the 2.5–97.5 percentiles, and dots are values lying outside the 2.5–97.5 percentiles.

**Figure 5 pathogens-12-00309-f005:**
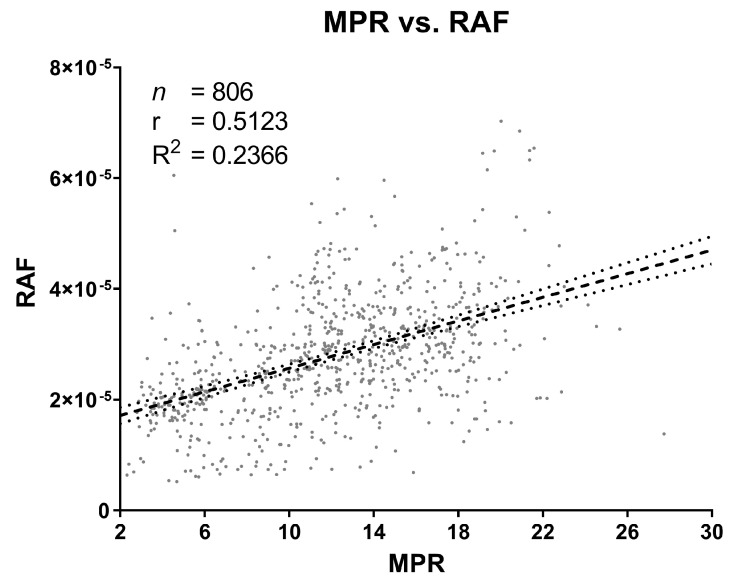
Correlation between maxpoint ratios (MPR) and their corresponding rates of amyloid formation (RAF; i.e., the relationship between signal intensity and reaction efficiency). While there seems to be a slight positive linear relationship, a large amount of scatter and weak correlation suggest that MPR values and their corresponding RAF values are independent of each other.

**Figure 6 pathogens-12-00309-f006:**
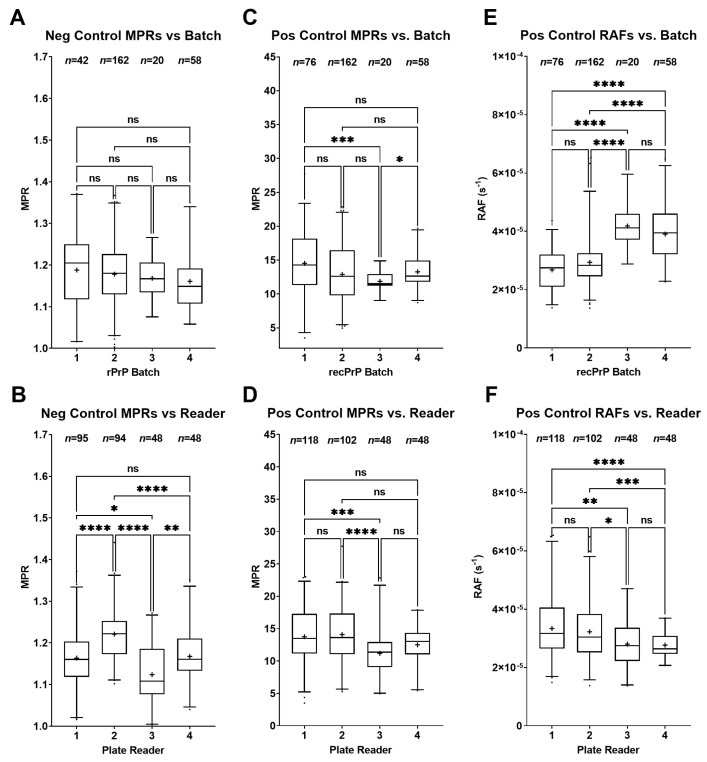
Comparisons of controls depending on the plate reader or recombinant substrate (recPrP) batch used. Boxes represent the interquartile range, lines are set at the median, “+” denotes the mean, and whiskers are the 2.5–97.5 percentiles (****, *p* < 0.0001; ***, *p* < 0.001; **, *p* < 0.01; *, *p* < 0.05; ns, *p* > 0.05): (**A**) no significant difference detected in negative-control maxpoint ratios (MPRs) between batches; (**B**) significant differences in negative-control MPRs between plate readers for every comparison except between plate readers 1 vs. 4; (**C**) significant differences between positive-control MPRs in batches 1 vs. 3 and 3 vs. 4; (**D**) significant differences in positive-control MPRs between plate readers 1 vs. 3 and 2 vs. 3; (**E**) significant difference was observed in the amyloid formation rates (RAF) of the positive controls between the first two batches (1 and 2) and the last two batches (1 and 3); (**F**) significant differences in RAFs were observed between all readers except 1 vs. 2 and 3 vs. 4.

**Figure 7 pathogens-12-00309-f007:**
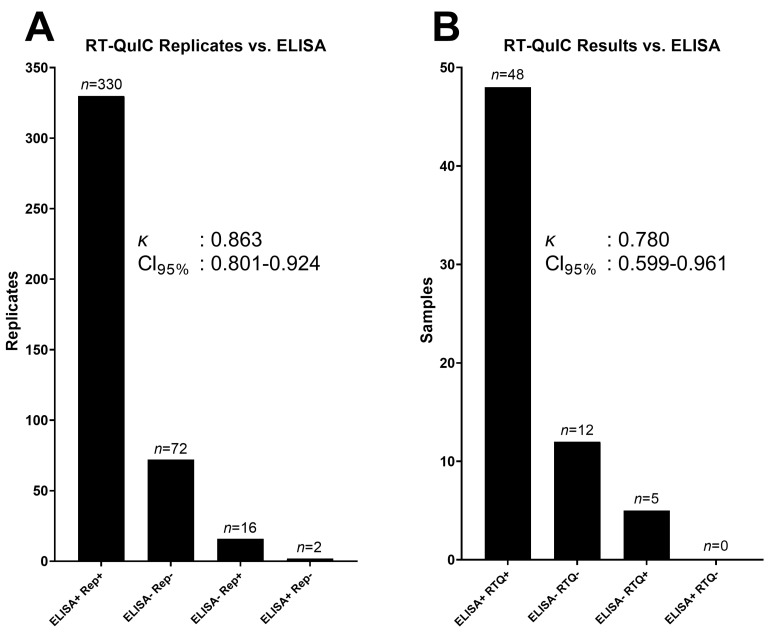
(**A**) Replicates that crossed the maxpoint ratio threshold (T${}_{MPR}$) compared with ELISA results. Of the 420 replicates, 402 agreed with the enzyme-linked immunosorbent assay (ELISA) results. Sixteen replicates crossed T${}_{MPR}$, but the sample was not detected by ELISA, and two replicates did not cross T${}_{MPR}$, but the sample was detected by ELISA; (**B**) after statistical analysis, 60 samples agreed with the ELISA results, 5 were positive for real-time quaking-induced conversion (RT-QuIC) but negative for ELISA, and none were negative for RT-QuIC but were positive for ELISA.

**Table 1 pathogens-12-00309-t001:** A compilation of the various methods used in publications for determining an RT-QuIC-positive sample when multiple replicates are processed. Positivity is typically assessed by a predetermined number of replicates crossing a study-specific threshold that is variably defined between 3 and 30 standard deviations from the initial mean relative fluorescent units (RFUs) of the entire plate. CWD: chronic wasting disease; CJD: Creutzfeldt–Jakob disease; PD: Parkinson’s disease; AD: Alzheimer’s disease; BSE: bovine spongiform encephalopathy; FFI: fatal familial insomnia; Sc: scrapie; CSF: cerebrospinal fluid; OM: olfactory mucosa; LN: lymph node; RAMALT: rectoanal mucosa-associated lymphoid tissue; stdev: standard deviations.

Threshold Used	Positivity Determination	Disease	Sample Type	References
20 stdev	Mann–Whitney u-test	AD	Brain	[52]
ine 10 stdev	50% of replicates	BSE	CSF	[21,26]
10 stdev	25% of replicates	BSE	Brain	[7]
ine 20 stdev	50% of replicates	CJD	Skin	[39]
10 stdev+10%	50% of replicates	CJD	CSF	[5]
10 stdev	50% of replicates	CJD	CSF; OM	[17,34,40,41,46]
5 stdev	50% of replicates	CJD	Brain	[43]
5 stdev	40% of replicates	CJD	CSF	[44]
3 stdev	50% of replicates	CJD	CSF	[22,42]
>10,000 RFU	50% of replicates	CJD	CSF	[48]
ine 10 stdev	Mann–Whitney u-test	CWD	Muscle; LN	[1,18]
10 stdev	2/3 of replicates	CWD	Blood; Feces; RAMALT	[19]
10 stdev	50% of replicates	CWD	Ticks; Feces	[36,57]
10 stdev	25% of replicates	CWD	Brain	[56]
10 stdev	1/3 of replicates	CWD	RAMALT	[23,30]
10 stdev	Avg of reps >T${}_{stdev}$	CWD	RAMALT	[28,29]
5 stdev	Mann–Whitney or Wilcoxon signed rank	CWD	Buffy-coat	[53]
5 stdev	Mann–Whitney u-test	CWD	Brain	[20,27,55]
5 stdev	Fisher’s exact test	CWD	Feces	[25]
5 stdev	Unpaired *t*-test	CWD	LN; Brain	[37]
5 stdev	One sample *t*-test	CWD	LN; Brain	[32]
5 stdev	50% of replicates	CWD	Feces	[31]
5 stdev	Avg of reps >T${}_{stdev}$	CWD	Brain; LN	[33,35]
4 stdev	50% of replicates	CWD	Blood	[49]
max signal of pos ctrl	1/3 of replicates	CWD	Brain	[24]
ine 5 stdev	Mann–Whitney or Wilcoxon signed rank	CWD; BSE	Brain	[38]
5 stdev	Avg of reps >T${}_{stdev}$	CWD; BSE	Brain	[54]
ine 3 stdev	50% of replicates	FFI	Brain	[47]
ine 30 stdev	50% of replicates	PD	CSF	[45]
10 stdev+10%	50% of replicates	PD	Brain	[3,50]
>120 RFU	50% of replicates	PD	Brain	[51]
ine 10 stdev	50% of replicates	Sc	Brain	[6]

## Data Availability

All data generated for this study are included in the manuscript and/or supplementary material.

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
