# Peer review of "Standardization of Data Analysis for RT-QuIC-Based Detection of Chronic Wasting Disease"

_pathogens, 2023, doi:10.3390/pathogens12020309_

Round 1
Reviewer 1 Report
In this paper, Rowden et. al. attempt to standardize the method for analyzing RT-QuIC data by establishing a TMPR threshold by which samples can then be normalized against between experiments. The manuscript is well written. While the need to standardize the RT-QuIC assay for the use in diagnostics is important, this paper falls well short with the methods provided. This summary will only include major points for consideration.
Major Points:
1. Overall method: What is being proposed in this paper appears to be no different than the method of standard deviations and reaction rates currently being used by other researchers in the field. The authors continually state that arbitrary thresholds (3-30 std dev) are being set by others yet a TMPR threshold of “2” is provided. This is contradicted in the discussion when stated that each researcher should determine their own value, thereby making the “2” arbitrary. Moreover, the RAF value that is being calculated from the MPR can be grossly misleading. The MPR can occur any time throughout the run and therefore can greatly skew the kinetic analysis. Additionally, the authors downplay the use of statistics against negative controls (line 48-51) yet generated statistics comparing unknown sample to negative controls.
2. Lack of negative controls/replicates: There were 668 unknown lymph nodes assessed within the study and only 2 negative controls were used. The variability between samples requires that a much larger cohort of negative animals should be used to validate the methods. This too should be carried over into the total number of replicates that are run for each sample. Four replicates on one plate is insufficient to obtain a reliable outcome for each sample.
3. Tissues used: While multiple lymph nodes were obtained and analyzed, the RPLN was excluded from the analysis. As this is one of the two main tissues used to diagnose CWD, establishing a standardization method based on other tissues undermines the purpose of this work. The differences in kinetics of individual tissues can be large but was not addressed. Bloodgood et. al (2021) has shown the detection differences in paired RPLN’s of the same animals. Furthermore, brain tissue generates much higher MPR’s than other tissues and one must consider tissue make-up and prion load when running the assay. This should also be addressed when comparing to the ELISA assay. The authors state several times that RT-QuIC is comparable or slightly more sensitive than ELISA with lymph nodes but others (McNulty in 2019 and Henderson in 2020) demonstrated that RT-QuIC can be up to 100,000 times more sensitive.
4. Address the variabilities in the RT-QuIC assay: The authors state that the variability between just 4 readers and 4 protein batches are statistically different. While the age of the BMG readers and the gain settings were brought up, other considerations for each machine such as correct filters and temperature also need to be addressed. Many organizations run at 55 degrees, have gain settings at 1700 or higher, and BMG filters are not always 450/480 – they send 448/482. All of these have dramatic effects on the kinetics of how the assay runs.
Author Response
We have included our responses in a PDF.

Reviewer 2 Report
This manuscript from Rowden et al. observed some results from RT-QuIC CWD literature and noticed a heterogeneity of methodology employed with a resulting inter-laboratory comparison of the data hard to make and biased, suspecptible to researchers preference. With a previously published algorithm approach (MPR), the authors aim to homogeneise the data through normalisation thus creating the possiblity of a better interpretation and comparison of results across different plates and laboratories.
The authors proceed into analysing CWD lymph nodes by RT-QuIC and esthablishing an optimal MPR, after comparing them with true diagnosis. The cut-off is then tested via comparison of the obtained MPR values and a standard CWD assay ELISA showing good concordance by ROC curve. There was no difference between the area where the lymph nodes were harvested but noticeably there was occasional machine reader and protein batches differences resulting in variable RT-QuIC results reliability.
Review: This article addressed a known issue in current CWD diagnosis and in more general RT-QuIC methodology around the interpretation and fidelity of results. Despite the topic being relevant to a relative small part of the neuroscience community, the unbaised mathematical approach in a significant step forward towards resolving the issue proposed but also shared with other more prominent diseases like CJD, tau- and a-synucleopaties, all using RT-QuIC as potential diagnostic tool. A reliable standardisation of RT-QuIC results is imperative if this tool is projected to replace other more expensive diagnostic methodology such as ELISA or post-mortem.
Upon a quick pubmed search with the criteria provided (keywords: RT-QuIC and "real-time quaking-induced 71 conversion"; between years 2012 and 2021) there are more than 200 hits. The reason why only 46 were analysed in unclear in the manuscript.
Some of the articles presented in Table 1 use different specimen sample (e.g lymph nodes vs CSF) but there is no comment on whether some articles use a certain criteria or another due the nature of the sample used.
Unfortunately, the authors do not comment on the ability to esthablish a correlation between MPR results and seeding quantity or disease duration to tease out further details about CWD that the algorithm could provide.
Nonetheless, this is a robust attempt to standardise RT-QuIC data in a prion pathology. In the research design, each experiment was weighted properly with relative controls and analysed under different aspects and methods, providing congruent and satisfactionary interpretation of the data. The statistical analysis is appropriate.
Specific comments : Line 20, provide reference
Author Response
We have included our response in a PDF.

Round 2
Reviewer 1 Report
In this paper, Rowden et. al. attempt to standardize the method for analyzing RT-QuIC data by establishing a TMPR threshold by which samples can then be normalized against between experiments. The manuscript is well written and there is a mutual agreement that the need to standardize the RT-QuIC assay for the use in diagnostics is important. That said, multiple facets of this paper still need to be addressed.
Major Points:
1. Thank you to the authors for providing clarification to my first original point. Please find additional comments below that explaining my position further.
Authors response:
If the std dev method were used, it would be impossible to determine a constant
threshold, because the std dev and average fluorescence between any two plates will
virtually always be different. This makes it difficult to reliably compare diagnostic results
between plates. On the other hand, our threshold (Tmpr) is only being used to calculate
the rate of the reaction, and diagnosis is determined rather by running a statistical test
against the MPR values of a known negative lymph node sample.
Reviewer response:
The purpose of determining individual thresholds between plates is because they are always different, there is no constant threshold. This is demonstrated even within your experiments showing that different readers and protein batches create statistically different results. The analysis of data though is not between the different plates but the samples that are being assayed and the controls they are being run against. There can be considerable variability within a sample/protein/reader, as pointed out in your discussion, which is why running 2 sets of multiple replicates on separate plates aids in confirming a positive or negative result. Here, each sample was run with 4 replicates but only on one plate.
1a. My suggestion would be to run each sample on at least 2 plates and possibly on different readers to validate your method. See point 2 for additional comments on negative controls.
From the authors response, obtaining a diagnosis by their method is different than the standard deviation method because a positive sample is determined by statistically comparing that samples MPR and the MPR of known negative controls. While the authors are correct that this is a different method, the premise is the same as those using standard deviations. In other studies, reaction rates from unknown samples are calculated based on a generated threshold, then statistically compared to the negative controls (please see references 20, 37, 53, 55). The difference in running an MPR analysis is that your negative controls will never have a value of zero, therefore you can run ANOVA to determine the significance. Basically, you could do the same thing with lag phase with the standard deviation model. The authors have taken substantial liberties in stating that the RAF’s of the negative controls in all the standard deviation experiments are ideal and always zero, essentially making the statistical analysis uninformative (L48-53 & L168-17). This is incorrect as there are false positive replicates within almost all the negative controls, which allows for statistics to be run. Again, there can be considerable variability within a sample and QuIC plates (your discussion). Additionally, this method introduces the problem where a sample could be more negative than the negative control and statistically significant. Please see point 2 for additional comments on negative controls.
1b. Were there any instances in the analyzed data where a statical significance was seen from a sample showing a lower MPR than the negative control?
Authors response:
The Tmpr threshold of "2" is based on the observed distribution of the empirical data.
Thus, this is not an arbitrary threshold. The threshold selection was supported by the
ROC analysis (Figure 3) when comparing MPR values to tissue-matched ELISA results. We
rephrase this in the manuscript for clarification in the results and methods (L106-112
and L302-304).
I thank the authors for providing further clarification to this point. Arbitrary was indeed the incorrect term for the RMPR threshold as the ROC and AUC data was clearly demonstrated in the distribution and empirical data. However, it appears that only one negative and positive control sample was used to validate these data points and the origin of that sample is not stated (additional questions in point 2). Therefore:
1c. What tissue was used as the positive and negative controls for these analyses?
1d. Could the authors please elaborate on how the adjustment of the RMPR threshold due to a multitude of variables (L213-217) should be done by researchers is different than the adjustment or determination of the standard deviations in previous studies (L40)? Are the authors suggesting that a ROC and AUC be performed on each tissue type to best determine the RMPR threshold? If not performed, would the selection of the RMPR threshold then not favor researcher bias too?
2. Lack of negative controls/replicates/samples:
Authors response:
The reviewer’s statement is inaccurate. All lymph nodes were independently
tested using ELISA. However, at the time of RT-QuIC testing, our team was blinded to the
diagnostic state of each sample, and we were unblinded after the RT-QuIC testing was
completed.
Reviewers response:
The testing of the lymph nodes by ELISA is not being questioned. The selection of samples to be run by RT-QuIC requires clarification as does the consistency of replicate and samples used for each analysis.
2a. Of the 533 WTD that were tested, how many tested positive by ELISA?
2b. Please provide the tissue type breakdown of the 4161 replicates used (L87).
2c. Of the 524 replicates that were positive, please provide the tissue type breakdown.
2d. For Figures 1 & 2, was only 1 negative and 1 positive sample used to generate the data?
2e. What tissue sample was used as the negative and positive control?
2f. Please provide graphs of the negative control MPRs and RAFs in Figure 2.
2g. What selection criteria was used for the additional 60 samples analyzed by RT-QuIC & ELISA?
Authors response:
We believe that the wording surrounding our experimental design was ambiguous, so
we added clarification in L291. We had many replicates of positive and negative controls
(48 plates; 285 replicates of neg control; 316 replicates of pos control; see Figure 2), and
four replicates for each blinded sample per plate (as is well-established standard
procedure for RT-QuIC).
Reviewers response:
The replicate number for the controls is not being questioned but the number of negative and positive control samples and the tissue types used for comparison is an important factor when running analysis. If only a single control from the brain is used throughout the experiments, comparisons to any lymph nodes may be inaccurate. The authors again demonstrated this with just their controls on their 4 plate readers and protein batches that significant differences occurred.
2h. Please see 1d. and 2e above. If only a single tissue was used, a suggestion of multiple tissue matched controls and separate analysis be done to validate results. If proof of principle for this method is what the authors are going for, this needs to be implicitly stated with the caveots addressed in the discussion.
2i. While 4 replicates per plate is well-established, the additional plate and replicates is also well established in confirming and validating the reproducibility and results of a given sample. Highly recommended that samples be run a second time with tissue matched controls.
Authors response:
Given that this was an opportunistic, retrospective analysis of data that was already
produced, it was outside the scope of this paper to assess the sensitivity of the number
of replicates per sample. For almost every plate, we ran six replicates per plate of the
same negative control so that there was consistency in the statistical analyses between
plates. When we compiled the MPR values, we observed that there were two distinct
distributions indicated in Figure 1. ROC analysis against ELISA results confirms that those two distributions correlate with ELISA diagnostic results.
Reviewers response:
2j. Could the authors please clarify why assessing the sensitivity of the number of replicates is outside the scope of this paper? Additionally, if this is not a concern, why were the RPLN, PPLN, and PSLN excluded from analysis for insufficient data points?
2k. By selecting a single control sample from a single tissue, the authors may have created an artificial distribution to analyze the other tissues. Furthermore, the ROC analysis against the ELISA could also be misrepresented based on the tissues selected (see 2g). Related to that, could the authors please clarify the status of all replicates (L148)? 528 replicates were produced with 402 agreeing with ELISA, 18 disagreeing – what about the other 108 replicates?
3. Tissues used:
Authors response:
Due to MNDNR practices for regulatory CWD testing, all RPLNs that tested
negative for ELISA were disposed of before we had access to them. Nevertheless, the
parotid lymph node is an appropriate tissue type for investigating CWD status via
RT-QuIC as documented by Schwabenlander, et al. 2020. We added a clarification in
L265-266. This is why we initially chose the parotid lymph node because every animal
had this tissue available. When we compared to ELISA, we ensured that all results were
tissue-matched (e.g. bilateral parotid RT-QuIC results vs. bilateral parotid ELISA results;
L109).
Reviewers response:
Thank you for the clarifications. This is indeed unfortunate that the negatives were disposed of, seeing the RPLN and obex are the only 2 tissues approved for CWD diagnosis by ELISA.
3a. Of the 515 parotid lymph nodes that were run by RT-QuIC, how many were run by ELISA?
3b. Were parotid negative and positive control tissues run?
3c. Please refer to response 2g; were the 60 samples selected for RT-QuIC and ELISA analysis the same as in the Schwabenlander paper? Could you please provide a breakdown of the positive and negative animals and tissue for that selection?
Authors response:
Concerning the sensitivity, this study addressed the analytical sensitivity and not the
assay sensitivity. While RT-QuIC does detect CWD at much lower dilutions, all animals in
this study were tested at the same dilution (L289-290). Moreover, the work by
Schwabenlander, et al from wild WTD populations showed great concordance between
RT-QuIC and ELISA results.
While it is understood that this study is investigating the analytical sensitivity of their method, overlooking the relevance of the assay sensitivity is a huge pitfall. The dilution factor of the samples is not in question. The controls, if not tissue matched, could possess bias as was stated in the first response: “brain tissue generates much higher MPR’s than other tissues and one must consider tissue make-up and prion load when running the assay”. RT-QuIC can detect prions at a significantly lower level than ELISA. If the 60 samples that were selected for RT-QuIC/ELISA comparison were all from animals with positive RPLN, the concordance between the 2 should be high as the RPLN is usually the first positive lymph node in CWD progression. The impact and validity of this study should come from samples that are ELISA negative and RT-QuIC positive. These are the samples that should have lower MPRs and RAFs, which may fall into that grey area of positive or negative based on the results and statistics run in comparison to the negative controls.
3d. Could the authors please include any additional data looking at samples that were ELISA negative and RT-QuIC positive? Could the authors also provide the matching data on more animals that were ELISA & RT-QuIC negative?
4. Address the variabilities in the RT-QuIC assay:
Reviewers response:
Thank you for the clarifying lines in the discussion.
4a. Could the authors please include the temperature and the gain settings in the material and methods section?
4b. Could the authors please provide the reason that a 48 hour cutoff was used to analyze the data?
Minor point:
In performing a literature search in PubMed on [RT-QuIC] and [Real-time quaking induced conversion] between 2012-2021, over 200 references were produced. Could the authors explain if there were additional criteria that they used to narrow down their results to 46?
Round 3
Reviewer 1 Report
**For this round of reviewer comments, only the last author response will be stated with the reviewer response after. Please refer to previous reviewer responses for clarification if needed. Additional statements will be included as addressed.
Authors’ response:
The positive and negative controls were performed on multiple plates/readers
aiming to understand the variability and reproducibility in results of the controls.
This helped in identifying an MPR threshold in which Sensitivity and Specificity
were maximum and that is able to include all potential variability in control
results observed by sample/reader/protein. We refer the reviewer to Figure 2
which shows that when all positive (and negative) controls are plotted
(regardless of rPrP batch or plate reader), MPRs and RAFs showed normal
distributions that never include the MPR value of 2 or less for any positive
controls.
Reviewer response:
While the authors clearly show the reproducibility of a single negative and single positive parotid lymph node as controls, the normal distribution of MPRs and RAFs still shows the degree of variability in single samples that can occur. The point being, analyzing only 4 replicates of unknown samples on one plate may not accurately represent that sample. Single plate analysis was prevalent in previous studies, but currently more and more labs are going to multiple plate replicate analysis to compensate for this variability.
Question: Are the authors aware of the method that the USDA is using to validate QuIC?
There was also no significant difference between batches for the negative
control (see Figure 6A) or negative tissue samples (see Figure 4). And, while our
analysis did show significant differences between readers for the negative
control (Figure 6B), these distributions are still well under the defined MPR
threshold of two. We reiterate that our selection of the MPR threshold took into
account potential significant variation due to readers/PrP batch.
All in all, the described method and threshold proposed account for potential
variability across readers, samples, and proteins, while proposing a
straightforward approach that is comparable across labs.
Reviewer response:
The statement is understood but should be written to more accurately represent the parotid lymph node as the samples this method is being applied too. And while the correlation between PLN, SMLN, and TLN, in Schwabenlander, et al. paper is understood, approximately 70% of the replicates run in this paper are from parotid lymph nodes.
Suggestion: Drop RPLN, PPLN, PSLN, obex, cerebrum, and cerebellum replicates from analysis as their purpose isn’t clear as most are excluded in comparison figures throughout the manuscript. State that the RPLN sample was used only as the diagnostic standard. As there are no negative control samples to include in the analysis, this makes it tough to use, even as a proof of concept sample.
Authors’ response:
The issue with performing a statistical test against a sample with zero variance is
that the actual rate values of unknown samples are inconsequential to the test.
A statistical test, such as ANOVA, will produce an exact p-value not based on the
values themselves, but rather based on how many values were not zero. This is
essentially equivalent to researchers predetermining how many replicates
should cross a given threshold to determine a diagnosis, and it is certainly the
case in the references the reviewer provided. We updated the wording on L51 to
include the reviewers point.
It is true that an ANOVA could also be performed on lag phase given that the
reaction was allowed to go 96+ hours or until all negative controls showed
positivity. However, for routine diagnostic purposes this is not feasible and is
unnecessary. While we did have some replicates of our negative control cross
TMPR, it was 4 out of 285 replicates (see Figure 2B).
Reviewer response:
Agreed and is why the suggestion of lag phase was proposed. The time the assay is allowed to run is subject to the researcher’s preference. The authors have selected a 48 hour cutoff for their analysis which could easily be applied to any QuIC assay being run.
Authors’ response:
We thank the reviewer for this comment. However, the usage of standardized
negative and positive controls is routine and expected for all diagnostic assays.
The performance of such controls across the multitude of experimental runs
helps to clearly define the observed variance. We have added the origin of the
control samples in the Materials and Methods section (L270-271).
Reviewer response:
Agreed. Thank you for adding the origin of the samples to M&M.
Because of this, please see previous comment (2 questions ago) on dropping miscellaneous samples and consider focusing on only PLN, SMLN, and TLN.
Authors’ response:
The adjustment is different because researchers would be determining a
constant threshold across all plates for determining RAF rather than calculating a
new one for every plate. If a sample’s average MPR is significantly larger from
the negative control, we consider that a positive. Based on the observations
reported herein, as well as ongoing research within our labs, we strongly believe
that the MPR will be a useful diagnostic tool for CWD. We support any
tangential analyses that researchers would like to perform in addition to MPR to
help determine diagnostic status.
Reviewer response:
In the authors opinion, how many control replicates would have to be run to establish an independent MPR for each specific tissue? From that, if this is to be interpreted correctly, the authors are suggesting each research lab establish an independent MPR for each sample they are analyzing, under the RT-QuIC conditions they are running in their lab? If this is the case, this should be stated in the introduction and not late in discussion.
Authors’ response:
Thirteen animals initially tested positive in RPLNs by ELISA (we added this
information on L270). The justification for the sub-selection of samples used for
RT-QuIC is provided in our previous publication, Schwabenlander et al. (2021).
For samples testing positive by RT-QuIC, a confirmatory ELISA was performed
which added one additional ELISA-positive animal on parotid LN but not on
RPLN.
The Minnesota Department of Natural Resources was aware of the 13 deer that
initially tested positive by RPLN ELISA, so those were included after the initial
PLN RT-QuIC sweep of 519 deer. The DNR then randomly selected the other 47
animals to be tested by RT-QuIC on the other sample types. Our team was
blinded during this process. After testing the various samples by RT-QuIC, our
team identified an additional four possible CWD-positive deer. ELISA was then
performed on all of the available lymph tissues from those 17 RT-QuIC-positive
animals (see Table 1 of the Schwabenlander, et al. [2021]).
Reviewers response to 2a-2g:
Thank you for providing a breakdown of the overall replicates as well as the replicates that were positive. Based on the data provided, this reviewer again suggests limiting the scope of this manuscript to PLN, SMLN, and TLN as this builds a much stronger case for the methods being presented.
Authors’ response:
We certainly agree with the reviewer that tissue-matched controls are
valuable to reliably analyze a sample using statistics. In this case
however, we are confident that comparing to the parotid negative
control was valid. This is due to the observation that for negative
samples of parotid, submandibular, and tonsil lymph nodes, there were
no significant differences in their MPR distributions (see Figure 4). We
added this caveat in L183-188.
Reviewers response:
Thank you to the authors for adding clarity. As the authors agree on tissue matched controls, previous comments on limiting the scope of the manuscript to these tissues is more applicable.
Authors’ response:
We confirmed the reproducibility for a given sample with both the
negative and positive controls as shown in Figure 2 and Figure 6. Each
was run on every batch and plate reader. Further, given the distributions
shown in Figure 4, we are confident that having a tissue matched control
is not necessary for PLNs, SMLNs, and TLNs which comprise over 92% of
the data.
Reviewers response:
Tissue matched controls, when validated are essential. The authors provide support for using PLN, SMLN, and TLN as the basis for this manuscript as 92% of their samples are comprised in these 3 tissues.
Authors’ second response:
The vast majority of RT-QuIC articles assessing lymph nodes use three to
four replicates. Further, this study focused on 533 individual animals
with multiple tissues for many of those animals; so throughput and lab
resources had to be considered. Therefore, the decision to use four
replicates was both pragmatic and based on literature.
The RPLNs, PPLNs, and PSLNs were excluded only from the distribution
analysis of samples with MPRs <2 (see Figure 4), but not from the
analysis as a whole. For instance, for the RPLN samples that were in the
lower distribution, there were ELISA data available, so they were
considered in the ROC analysis. There were no available ELISA data for
PPLNs or PSLNs.
Reviewers response:
The reviewer understands the throughput and resources required to analyze the number of samples presented within. If the decision to include data was based on the available matched tissues with RT-QuIC and ELISA data as stated, then the analysis should contain that data.
Authors’ second response:
We state that control replicates were removed (L88-90) before any
analysis was done on the distributions for exactly that reason. Thus the
controls couldn’t have applied any bias on the results of any
downstream analyses (see Figure 1).
The ROC analysis did not take into consideration any data from the
negative control; only samples which were tested side-by-side with
RT-QuIC and ELISA. The two control samples could not have skewed the
data, because they were not included in the analysis.
We thank the reviewer for pointing out the discrepancy between the
numbers. We realized that “528” was a number we overlooked before
excluding tissues that were not tissue-matched exactly with ELISA
results. We have updated the manuscript to the correct “420” instead
(L149).
Reviewers response:
Thank you for clarifying the discrepancy with replicates. To expand upon the point of using matched tissues, only the tissues from the 17 (out of 60) animals that were RT-QuIC positive were submitted for ELISA. As previously stated, QuIC is more sensitive than ELISA so you would expect the correlation to be high. Were the other 43 animal tissues run by ELISA?
Authors’ response:
As mentioned above in our answer to 2g., 60 animals were selected for a
second round of RT-QuIC testing with PLNs, SMLNs, and TLNs. Not
mentioned in this study was that we also tested blood and feces, but
those sample types lied outside the scope of this study. During this
second round of testing, we identified 17 animals that had amyloid
seeding activity in various sample types. These 17 animals were then
tested by ELISA for each tissue type including the bilateral parotid.
Reviewers response:
Please see previous response.
Authors’ response:
They were the same samples as reported in Schwabenlander et al.
(2021). Thirteen animals tested positive for RPLN ELISA of the 519 total
from that study. The MN DNR (without un-blinding us) sub-selected 60
of those animals and made sure to include the thirteen known positives.
We then tested those 60 by PLN (again), SMLN, and TLN. As stated
before, we received only the ELISA-positive RPLNs after the blind was
broken which we tested by RT-QuIC.
We have attempted to clarify this information on L270 and L287-290.
Reviewers response:
Thank you for clarifying the response. Could the authors please state the reason for why they did not submit all 60 animal tissues for ELISA analysis?
Authors’ response:
We are unsure what the reviewer is quoting concerning the brain tissue MPRs.
We cannot say that brain tissue generates higher MPRs because we have not
investigated that hypothesis. The 60 animals were not all positive animals. Only
13 animals were positive by initial RPLN testing.
Reviewers response:
The quote was from this reviewer’s initial response to the authors. The point that was being made was that different tissue can generate different MPRs and RAFs, hence the need for tissue matched controls when trying to establish a diagnostic method.
Authors’ response:
We recommend the reviewer read Schwabenlander et al. (2021) wherein
comparisons between ELISA and RT-QuIC are discussed. The emerging consensus
in the scientific community is that RT-QuIC is more sensitive than ELISA. Thus a
negative ELISA and positive RT-QuIC result is not surprising. Out of 65 samples
there were 5 that were ELISA negative and RT-QuIC positive. Zero were ELISA
positive and RT-QuIC negative. We point the reviewer to Figures 3 and 7.
Reviewers response:
The reviewer has read Schwabenlander et al. (2021). The point is that only deer that had a positive RT-QuIC result were submitted for confirmatory ELISA. As QuIC is more sensitive than ELISA, the results from this experiment were expected. Furthermore, the authors should note that the sensitivity differences between lymph nodes and brain material using ELISA (Schwabenlander et al. 2021 vs. McNulty et. al. 2019) demonstrates a large difference.
Authors’ response:
We have updated L300-302 with this information.
A 48 hour cutoff was a pragmatic decision given the sample size and throughput
capabilities of our lab. Further, much of the RT-QuIC literature indicates a 48
hour runtime for lymph nodes considering the false positivity issues the
reviewer mentioned above at long runtimes.
Reviewers response:
Thank you to the authors for providing a temperature and gain setting.
Additionally, thank you for clarity on the timeframe used in the assay.
Authors’ response:
We also included the criteria that “studies were selected if the researchers both
performed RT-QuIC and implemented a rubric for determining TSE status.” This
narrowed the search considerably since many studies did not have a specific
methodology for determining a TSE diagnosis.
Reviewers response:
Thank you for providing clarity on this point
